# Loss of neurofibromin Ras-GAP activity enhances the formation of cardiac blood islands in murine embryos

Amanda D Yzaguirre[1,2†], Arun Padmanabhan[3,4†‡], Eric D de Groh[3,4†§], Kurt A Engleka[3,4†], Jun Li[3,4], Nancy A Speck[1,2*], Jonathan A Epstein[3,4*]

[1]Abramson Family Cancer Research Institute, Perelman School of Medicine at the University of Pennsylvania, Philadelphia, United States; [2]Department of Cell and Developmental Biology, Perelman School of Medicine at the University of Pennsylvania, Philadelphia, United States; [3]Institute for Regenerative Medicine, Perelman School of Medicine at the University of Pennsylvania, Philadelphia, United States; [4]Cardiovascular Institute, Perelman School of Medicine at the University of Pennsylvania, Philadelphia, United States

*For correspondence: nancyas@exchange.upenn.edu (NAS); epsteinj@upenn.edu (JAE)

[†]These authors contributed equally to this work

Present address: [‡]Department of Medicine, Massachusetts General Hospital, Harvard Medical School, Boston, United States; [§]Medpace Inc., Cincinnati, United States

Competing interests: The authors declare that no competing interests exist.

**Abstract** Type I neurofibromatosis (NF1) is caused by mutations in the *NF1* gene encoding neurofibromin. Neurofibromin exhibits Ras GTPase activating protein (Ras-GAP) activity that is thought to mediate cellular functions relevant to disease phenotypes. Loss of murine *Nf1* results in embryonic lethality due to heart defects, while mice with monoallelic loss of function mutations or with tissue-specific inactivation have been used to model NF1. Here, we characterize previously unappreciated phenotypes in *Nf1-/-* embryos, which are inhibition of hemogenic endothelial specification in the dorsal aorta, enhanced yolk sac hematopoiesis, and exuberant cardiac blood island formation. We show that a missense mutation engineered into the active site of the Ras-GAP domain is sufficient to reproduce ectopic blood island formation, cardiac defects, and overgrowth of neural crest-derived structures seen in *Nf1-/-*embryos. These findings demonstrate a role for Ras-GAP activity in suppressing the hemogenic potential of the heart and restricting growth of neural crest-derived tissues.

## Introduction

NF1 is a common human disorder characterized by benign and malignant tumors of neural crest origin, pigmentation defects, learning disorders, cardiovascular abnormalities and a wide spectrum of other abnormalities including a predilection for leukemia (especially juvenile myelomonocytic leukemia, [JMML]) and vascular defects (*Cichowski and Jacks, 2001*; *Friedman et al., 2002*). Some of these phenotypes, including JMML and vascular defects, are shared by patients with related disorders associated with activation of the Ras signaling pathway, which together have been termed the 'RAS-opathies' (*Rauen et al., 2010*). Neurofibromin contains a protein domain termed the GAP-related domain (GRD) that is homologous to yeast IRA proteins. The NF1 GRD is able to complement yeast IRA mutants and hydrolyze GTP bound to active Ras, thereby down-regulating Ras signaling (*Ballester et al., 1990*; *Xu et al., 1990*). Interestingly, however, missense mutations in humans with NF1 have been identified that alter amino acids throughout the protein, suggesting functional domains outside of the GRD (*Mattocks et al., 2004*). Additional cellular functions for neurofibromin have also been identified, including modulation of protein kinase A (PKA) and cyclic adenosine monophosphate (cAMP) pathways (*Brown et al., 2010*; *The et al., 1997*; *Wolman et al., 2014*). A C-terminal region of neurofibromin has also been shown to interact with a major class of

**eLife digest** Messages are carried from the surface of a cell to the cell's nucleus in order to regulate various processes such as how often the cell will divide. The Ras-signaling pathway carries some of these messages. A gene called *Nf1* encodes a protein in this pathway that deactivates another protein called Ras when the message is no longer required. If a mutation in *Nf1* prevents it from deactivating Ras, the pathway becomes hyperactivated. In humans, this results in a disorder called Neurofibromatosis type I, which is characterized by tumors that affect many parts of the body.

When the expression of *Nf1* is turned off in mice, the mice die as embryos because of cardiac defects. But a mouse in which *Nf1* has been turned off in specific organs or tissues other than the heart can survive, and these mice are used to model Neurofibromatosis type I and to help to identify potential treatments.

Yzaguirre et al. have now identified new roles for *Nf1* during embryonic development. In the embryo, blood cells originate from the cells lining the blood vessels. The experiments revealed that, when the *Nf1* gene was mutated in mice, fewer blood cells formed from the lining of the major blood vessel that leaves the embryonic heart. In contrast, these mutant mice formed more structures called cardiac blood islands than a normal mouse. These structures line the heart, and have the potential to generate new blood cells for the heart to pump. These results shed new light on how blood is originally formed from the lining of the heart and blood vessels, and show that Ras signaling must be tightly regulated to maintain normal blood development in the embryo.

Furthermore, Yzaguirre et al. demonstrated that this excessive formation of cardiac blood islands resulted specifically from the loss of *Nf1*'s role in the Ras-signaling pathway. This was achieved by using gene targeting to generate a mouse that expresses *Nf1* with a minor change that affects only the protein's interaction with Ras. In the future, this new strain of mouse will be a useful tool in determining if specific aspects of Neurofibromatosis type I can be attributed to loss of *Nf1*'s role in Ras-signaling and could therefore be treated by medicines that target this pathway.

heparan sulfate proteoglycans (*Hsueh et al., 2001*) while full-length neurofibromin can bind to the scaffolding domain of caveolin-1 (*Boyanapalli et al., 2006*). Therapeutic strategies for the treatment of NF1 have focused on modulation of the Ras pathway, but the degree to which Ras dysregulation accounts for the diverse aspects of the human disease, or for the equally diverse features of various animal models of NF1, remains a critical question in the field.

Mouse models of NF1 have demonstrated critical developmental functions for neurofibromin in multiple tissues, including neural crest, endothelium, and hematopoietic stem and progenitor cells (HSPCs) (*Brossier and Carroll, 2012*; *Costa and Silva, 2003*; *Gitler et al., 2003*; *Bollag et al., 1996*; *Zhang et al., 1998*). HSPCs arise during midgestation from a transient population of endothelial cells called hemogenic endothelium (HE) located in the yolk sac, the dorsal aorta, vitelline and umbilical arteries (*Bertrand et al., 2010*; *Boisset et al., 2010*; *Chen et al., 2009*; *Kissa and Herbomel, 2010*; *Lam et al., 2010*; *Oberlin et al., 2010*; *Zovein et al., 2008*). HE gives rise to HSPCs through a direct transition of endothelial cells into hematopoietic cells, independent of cell division (*Kissa and Herbomel, 2010*; *Eilken et al., 2009*). This endothelial to hematopoietic transition (EHT) was thought to occur only in the major arteries of the embryo, the placenta, and the yolk sac, but recent studies have identified the heart and the head as sites of *de novo* hematopoiesis (*Dzierzak and Speck, 2008*; *Nakano et al., 2013*; *Li et al., 2012*). In the heart, hemogenic endocardial cells are integrated into the outflow cushion and atria and undergo EHT on embryonic day (E) 9.5. Unlike arterial HE cells that give rise to the full repertoire of hematopoietic cells, hemogenic endocardial cells produce a transient population of hematopoietic cells restricted to the erythroid/myeloid lineage, similar in potential to an early wave of erythroid/myeloid progenitors (EMPs) that emerge starting at E8.5 in the yolk sac (*Nakano et al., 2013*; *Palis et al., 1999*).

Later in gestation, the heart is associated with a less-defined second wave of hematopoiesis characterized by aggregates of endothelial and hematopoietic cells called blood islands. Cardiac blood island formation is a prevalent physiological process that has been identified in embryonic mice, chicks, quails and humans, but surprisingly, little is known about the formation of these structures (*Hiruma and Hirakow, 1989*; *Hutchins et al., 1988*; *Kattan et al., 2004*; *Ratajska et al., 2006*;

*Red-Horse et al., 2010*; *Wu et al., 2013*; *Jankowska-Steifer et al., 2015*). What is known about cardiac blood islands comes primarily from histological studies. Blood islands form in the subepicardial space near the interventricular sulci between E11 and E14 and consist primarily of erythroblasts, but have also been associated with megakaryocytes, platelets, and leukocytes (*Ratajska et al., 2009*; *Red-Horse et al., 2010*). Clonal and histological analysis suggests that blood islands emerge from the endocardium, protruding into the myocardium where they pinch off, forming blood-filled spheres or tubules that join the coronary plexus (*Red-Horse et al., 2010*; *Jankowska-Steifer et al., 2015*). It has been suggested that hematopoietic cells enter cardiac blood islands by diapedesis, but other routes such as circulation or the de novo generation of hematopoietic cells from the endocardium in situ have not been ruled out (*Ratajska et al., 2006*). Cardiac blood island formation was found to be more robust in *Tbx18* null mouse embryos, and thought to be an indirect consequence of aberrant signaling (*Wu et al., 2013*). Here we show that hyperactive Ras signaling increases cardiac blood island formation, and that endocardial cells of the blood islands have functional characteristics of HE and express Runx1, a marker of HE.

## Results

### *Nf1* deficiency increases yolk sac hematopoiesis but decreases specification of hemogenic endothelium in the dorsal aorta

E 12.5–13.5 *Nf1*-deficient fetuses were reported to have increased numbers of committed hematopoietic progenitors in the fetal liver (*Largaespada et al., 1996*; *Bollag et al., 1996*; *Zhang et al., 1998*). Since many fetal liver progenitors in the midgestation embryo originate in the yolk sac (*Lux et al., 2008*), we examined the number of EMPs in the yolk sac of E10.5 *Nf1*-deficient embryos. *Nf1-/-* yolk sacs (*Figure 1A*) contained significantly more EMPs, specifically due to an increased number of erythroid progenitors (*Figure 1B*), suggesting that Ras signaling positively regulates EMP numbers. We next examined the impact of Nf1 deficiency on hematopoiesis in the major arteries. The majority of HE cells in the major arteries (dorsal aorta, vitelline and umbilical) undergo EHT between E9.0–10.5, resulting in the formation of $Kit^+$ $CD31^+$ $Runx1^+$ hematopoietic cells that remain briefly attached as clusters to the luminal wall of the arteries. In contrast to the increase in EMPs observed in the yolk sac, $CD31^+$ $Kit^+$ $Runx1^+$ hematopoietic cluster cells were decreased in the dorsal aortas of E10.5 *Nf1*-deficient embryos (*Figure 1C,D*). The decrease in $CD31^+$ $Kit^+$ $Runx1^+$ hematopoietic cluster cells appears to be due to decreased *de novo* generation, as fewer $Runx1^+$ $CD31^+$ $Kit^{-/low}$ HE cells were present in the dorsal aortas (*Figure 1C,D*). These data suggest that disruption of neurofibromin function augments the formation of EMPs in the yolk sac, but inhibits the specification of hemogenic endothelium in the dorsal aorta.

### Nf1 deficiency results in ectopic cardiac blood island formation

*Nf1* deficiency results in embryonic lethality by midgestation (approximately E13) due to cardiac defects. These defects include enlarged endocardial cushions and a malformed outflow tract (*Brannan et al., 1994*; *Jacks et al., 1994*; *Lakkis and Epstein, 1998*). Despite midgestation lethality, E11.5 *Nf1-/-* embryos appeared grossly normal (*Figure 2A*). However, blood -filled protrusions were often visible on the ventricles of *Nf1-/-* embryos (*Figure 2B* arrowheads). Whole-mount confocal analyses revealed that the protrusions are blood island-like structures budding from the ventricular endocardium, as they express CD31 and the hematopoietic markers CD41 and Runx1 (*Figure 2C* arrowheads). The blood islands are concentrated laterally on both ventricles of *Nf1-/-* embryos (*Figure 2C*), in contrast to wild-type embryos, in which it was reported that blood islands are generally located on the dorsal surface in the interventricular sulcus (*Jankowska-Steifer et al., 2015*). Cardiac blood island formation was more robust in *Nf1*-deficient embryos compared to $Nf1^{+/+}$ and $Nf1^{+/-}$ littermates; an average of $63.7 \pm 7.6$ blood islands could be identified via confocal microscopy on the ventricles of E11.5 *Nf1-/-* embryos, whereas $Nf1^{+/+}$ and $Nf1^{+/-}$ littermates averaged $0.3 \pm 0.6$ and $1.2 \pm 1.3$ blood islands, respectively (*Figure 2D*). At E10.5, only 25% (1/4) of *Nf1*-deficient embryos displayed small budding cardiac blood islands, whereas 92% (11/12) of E11.5 embryos had robust blood island formation, indicating that cardiac blood islands arise between E10.5 and 11.5 in *Nf1-/-* embryos.

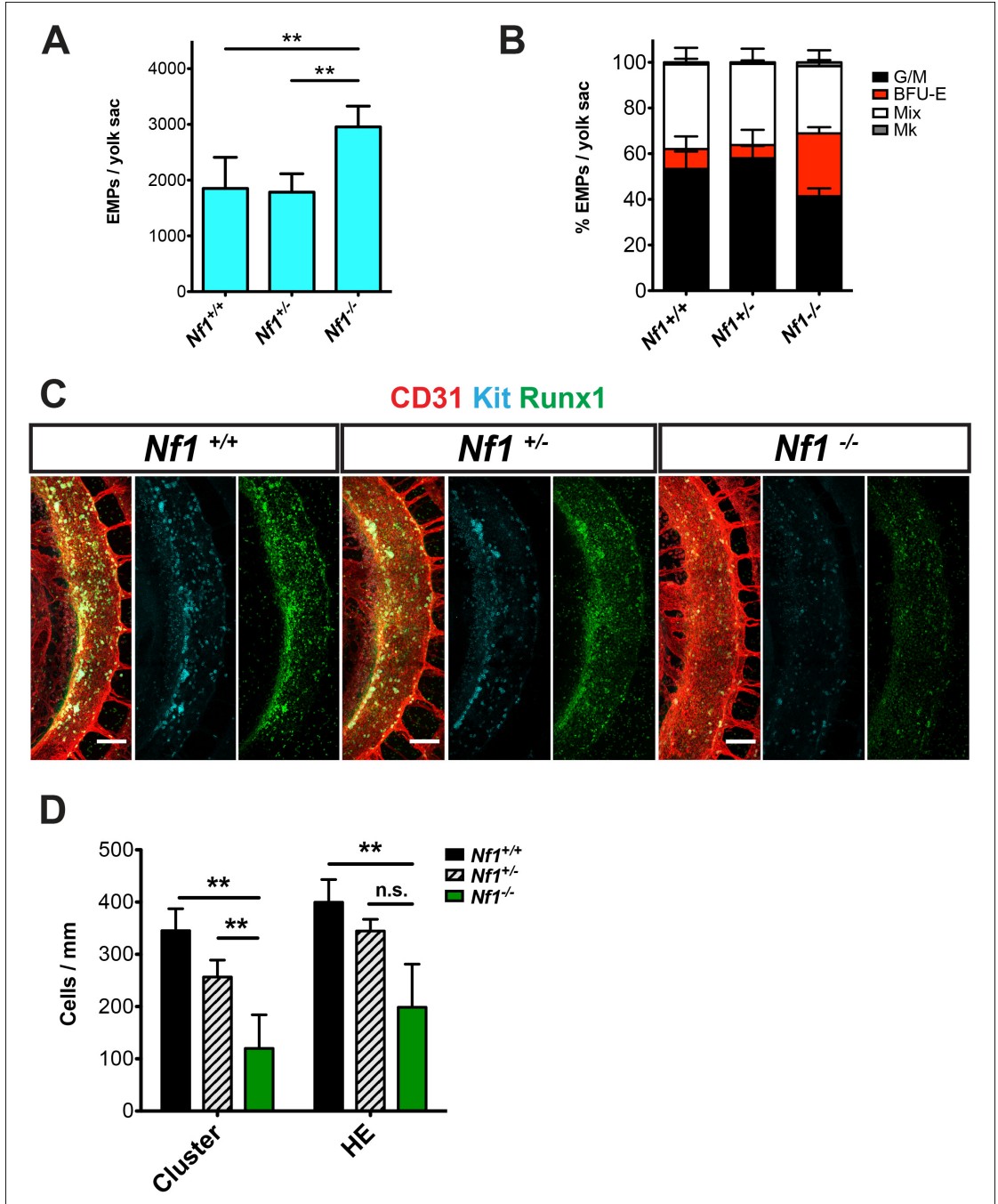

**Figure 1.** *Nf1* deficiency increases yolk sac hematopoiesis but decreases specification of hemogenic endothelium in the dorsal aorta at E10.5. (A) Quantification of erythroid and myeloid progenitors (EMPs) in the yolk sacs of E10.5 *Nf1⁺/⁺*, *Nf1⁺/⁻* and *Nf1-/-* conceptuses (*Nf1⁺/⁺* n = 7; *Nf1⁺/⁻* n = 8; *Nf1-/-* n = 3). One-way ANOVA and Bonferroni's multiple comparison test was applied to determine significance, error bars represent the standard deviation (SD) (B) Percent of EMP colony type. Mk: megakaryocyte; Mix: granulocyte-erythroid-monocyte-megokaryocyte; BFU-E: burst forming unit-erythroid; G/M: granulocyte-macrophage colonies. There were significantly more BFU-E progenitors in the yolk sacs of *Nf1-/-* compared to *Nf1⁺/⁺* and *Nf1⁺/⁻* littermates, p≤0.0001. (C) Confocal Z-projections (Z intervals = 2 μm) of *Nf1⁺/⁺* , *Nf1⁺/⁻* and *Nf1-/-* dorsal aortas at E10.5, immunostained for CD31 (red) Runx1 (green) and Kit (cyan). Scale bars = 100 μm. Aortas are oriented with the ventral aspect on the left. (D) Quantification of CD31⁺ Runx1⁺ Kit⁺ hematopoietic cluster cells and CD31⁺ Runx1⁺ Kit⁻/low hemogenic endothelial cells within the dorsal aorta at E10.5, One-way ANOVA and Bonferroni's multiple comparison test applied to determine significance, error bars represent the SD and n = 3 for all genotypes. ** indicates that p≤0.01.

To determine if the ectopic cardiac blood islands harbored functional hematopoietic progenitors we performed colony-forming assays. To eliminate circulating blood cells, the atrium was removed and circulating blood flushed from the ventricles before the ventricles were dissociated and plated in methylcellulose supplemented with cytokines. *Nf1-/-* ventricles contained significantly more EMPs than their *Nf1+/+* and *Nf1+/-* littermates (*Figure 2E*). This suggests that the phenotypic hematopoietic cells in the blood islands are functional erythroid and myeloid progenitors.

We used CD31, Runx1, Kit and CD41 whole-mount immunofluorescence and confocal microscopy to examine the structure of cardiac blood islands in *Nf1*-deficient ventricles at E11.5. Single optical projections through the blood islands indicate that they are cystic structures that consist of a layer of CD31$^+$ endocardial cells that is continuous with the endocardium lining the ventricular trabeculae (*Figure 3A,C*). A layer of 3–4 CD31 bright cells with morphology between a flat endocardial cell and a rounded hematopoietic cell lined the base of most blood islands (*Figure 3A–D*). Some of these cells express the hemogenic endothelial marker Runx1 but they do not express high levels of the early hematopoietic markers CD41 and Kit, suggesting that they are hemogenic endocardial cells that have not yet initiated the transition into hematopoietic cells (*Figure 3B,D*, arrows). Within the cystic structure of most blood islands, there are also rounded cells that are CD31$^+$ Kit$^+$ Runx1$^+$ or CD31$^+$ CD41$^+$ Runx1$^+$; these cells are phenotypic and morphological HSPCs (*Figure 3B,D*, arrowheads). These data suggest that blood islands are derived from the endocardium of the ventricle and that the endocardium of blood islands has a latent HE potential that is held in check by Ras-GAP activity.

In addition to robust cardiac blood island formation, E11.5 *Nf1-/-* embryos have enlarged fetal livers populated by Runx1$^+$ and CD41$^+$ hematopoietic cells (*Figure 2C*), consistent with previous studies that found significantly higher numbers of fetal liver clonogenic progenitors (*Zhang et al., 1998*; *Largaespada et al., 1996*; *Bollag et al., 1996*). Furthermore, competitive repopulation assays comparing Sca1$^+$lin$^{-/dim}$ cells isolated from the fetal livers of *Nf1-/-* and *Nf1+/+* embryos demonstrated that *Nf1-/-* cells have a growth advantage, particularly in the myeloid compartment (*Bollag et al., 1996*). Thus, the enlargement of the fetal liver may be due to elevated proliferation of *Nf1-/-* hematopoietic cells.

## Creation of *Nf1* R1276P GRD mice

In order to determine if the increase in cardiac blood islands seen in *Nf1-/-* embryos is due specifically to loss of the Ras-GAP activity of neurofibromin, we engineered a missense mutation within the GRD. Arginine 1276 has been shown to be the 'arginine finger' of the GRD and is critical for catalytic activity. Mutation of this residue to proline was identified in a family with NF1, and crystal structures of related GAP domains were consistent with empiric studies showing loss of GAP activity following R1276P mutagenesis (*Ahmadian et al., 1997*; *Scheffzek et al., 1997*; *Klose et al., 1998*; *Hiatt et al., 2004*). We generated 'knockin' mice in which arginine 1276 was mutated to proline (R1276P) and designated these mice *Nf1$^{GRD/+}$* (*Figure 4—figure supplement 1*). We generated an additional line of engineered mice in order to control for minor changes to intronic genomic sequences necessitated by the gene targeting and selection strategy (see Materials and methods and *Figure 4—figure supplement 1*). For these control mice, designated *Nf1$^{GRDCTL/+}$*, we utilized the identical targeting strategy but arginine 1276 was left intact. Appropriate targeting in several ES cell clones for each of the *Nf1$^{GRD}$* or *Nf1$^{GRDCTL}$* constructs was demonstrated by Southern blotting (*Figure 4—figure supplement 1*). These were used to generate chimeric animals that were then bred for germ line transmission.

Heterozygous *Nf1$^{GRD/+}$* mice appeared normal and were able to breed, but heterozygous intercrosses failed to produce any viable homozygous *Nf1$^{GRD/GRD}$* pups (*Table 1*). One out of 61 embryos genotyped at E12.5 was *Nf1$^{GRD/GRD}$*, and 11of 63 (17.5%) were *Nf1$^{GRD/GRD}$* at E11.5 (*Table 1*). Hence, homozygous R1276P mutation of *Nf1* causes midgestation embryonic lethality with most embryos succumbing by E12.5.

Total cell lysates from *Nf1$^{GRD/+}$* and *Nf1$^{GRD/GRD}$* embryos exhibited similar levels of neurofibromin protein of expected apparent molecular weight of 250–280 kDa (*Figure 4A*). The relative neurofibromin protein expression was similar to that of wild-type embryos and was increased relative to *Nf1+/-* embryos (*Figure 4B*).

To assess whether the introduced R1276P mutation within the GAP domain of neurofibromin disrupts Ras-GAP activity *in vivo*, tissues were examined for elevated phosphorylated extracellular-

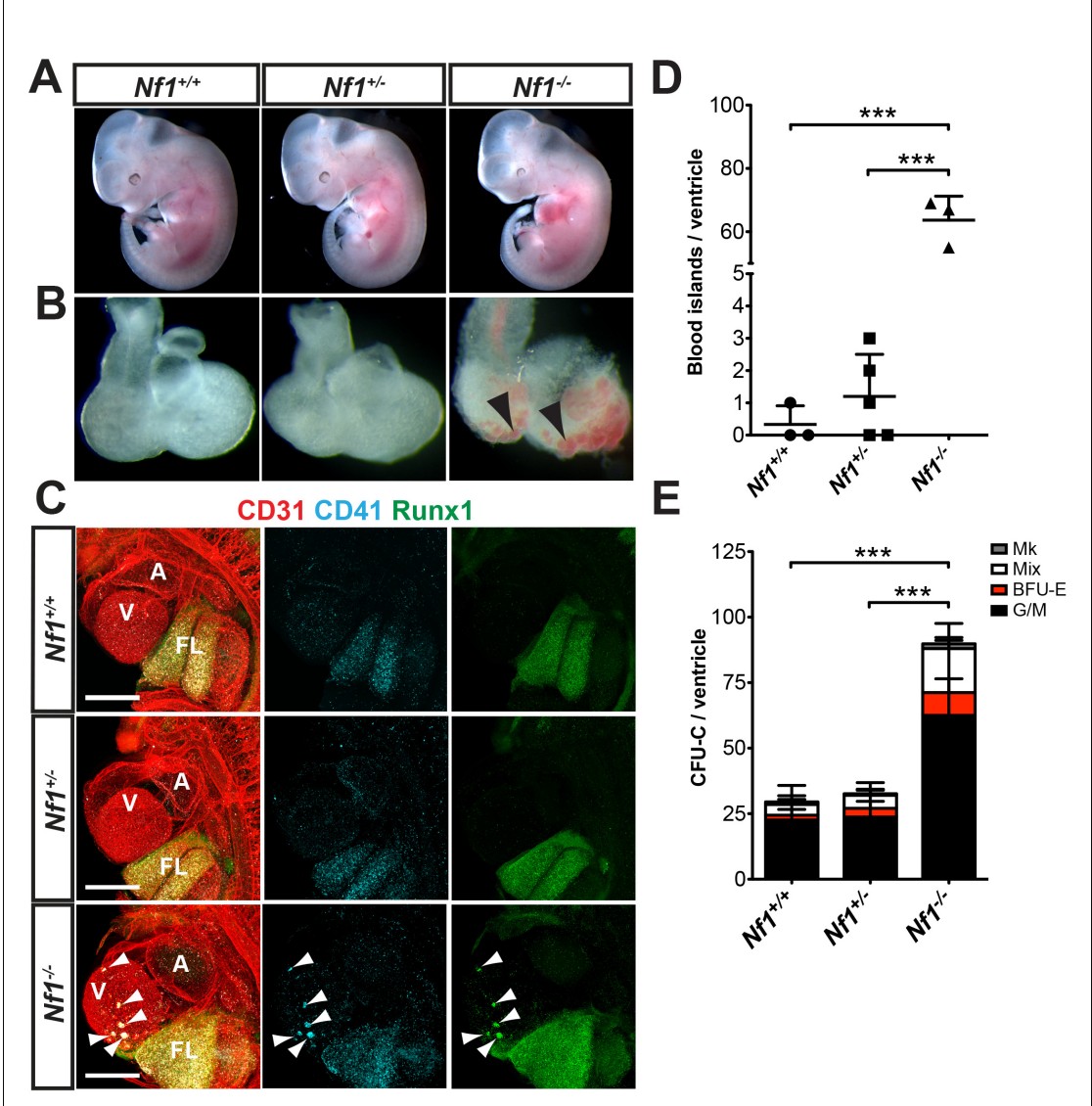

**Figure 2.** Ectopic formation of cardiac blood islands in *Nf1*-/- embryos. (**A**) Gross view of E11.5 *Nf1*+/+, *Nf1*+/- and *Nf1*-/- embryos (**B**) Isolated hearts from embryos in (**A**) Black arrowheads point to two examples of blood-filled protrusions. (**C**) Confocal Z-projections (Z interval = 5 µm) of CD31 (red), CD41 (cyan) and Runx1 (green) immunostained *Nf1*+/+, *Nf1*+/- and *Nf1*-/- E11.5 embryos. Blood island-like structures (arrowheads) are visible on the ventricle of the *Nf1*-/- embryo. Scale bars = 500 µm. (**D**) Quantification of blood islands on the ventricles of E11.5 embryos, One-way ANOVA and Bonferroni's multiple comparison test applied to determine significance, error bars represent SD. (**E**) Number of erythroid and myeloid progenitors per flushed E11.5 ventricles. One-way ANOVA and Bonferroni's multiple comparison test applied to determine significance, error bars represent the standard SD. *Nf1*+/+ n = 10, *Nf1*+/- n = 33, and *Nf1*-/- n = 6. \*\*\* indicates that p≤0.001. CFU-C: colony-forming units-culture; V: ventricle; A: atrium; FL: fetal liver; Mk: megakaryocyte; Mix: granulocyte-erythroid-monocyte-megakaryocyte; BFU-E: burst forming unit-erythroid; G/M: granulocyte-macrophage colonies.

signal regulated kinase (pERK), a downstream effector of Ras, as evidence of up-regulated Ras pathway activity. *Nf1*GRD/flox newborns in which *Nf1* was deleted by *Wnt1-Cre*, displayed elevated pERK staining in neural crest-derived tissues such as peripheral nerves (*Figure 4C*), within hyperplastic adrenal medullary tissue (*Figure 4D*), and in enteric ganglia (*Figure 4—figure supplement 2*). *Wnt1-Cre; Nf1*GRD/flox newborns showed prominent pERK staining in the axons and cell bodies of peripheral nerves (*Figure 4—figure supplement 2*) that was not observed in control animals. Elevated pERK staining was also seen in the enlarged cardiac cushions of *Nf1*GRD/GRD embryos, indicating the R1276 mutation is sufficient to elevate pERK levels (*Figure 4E*). Multiple reports showed that

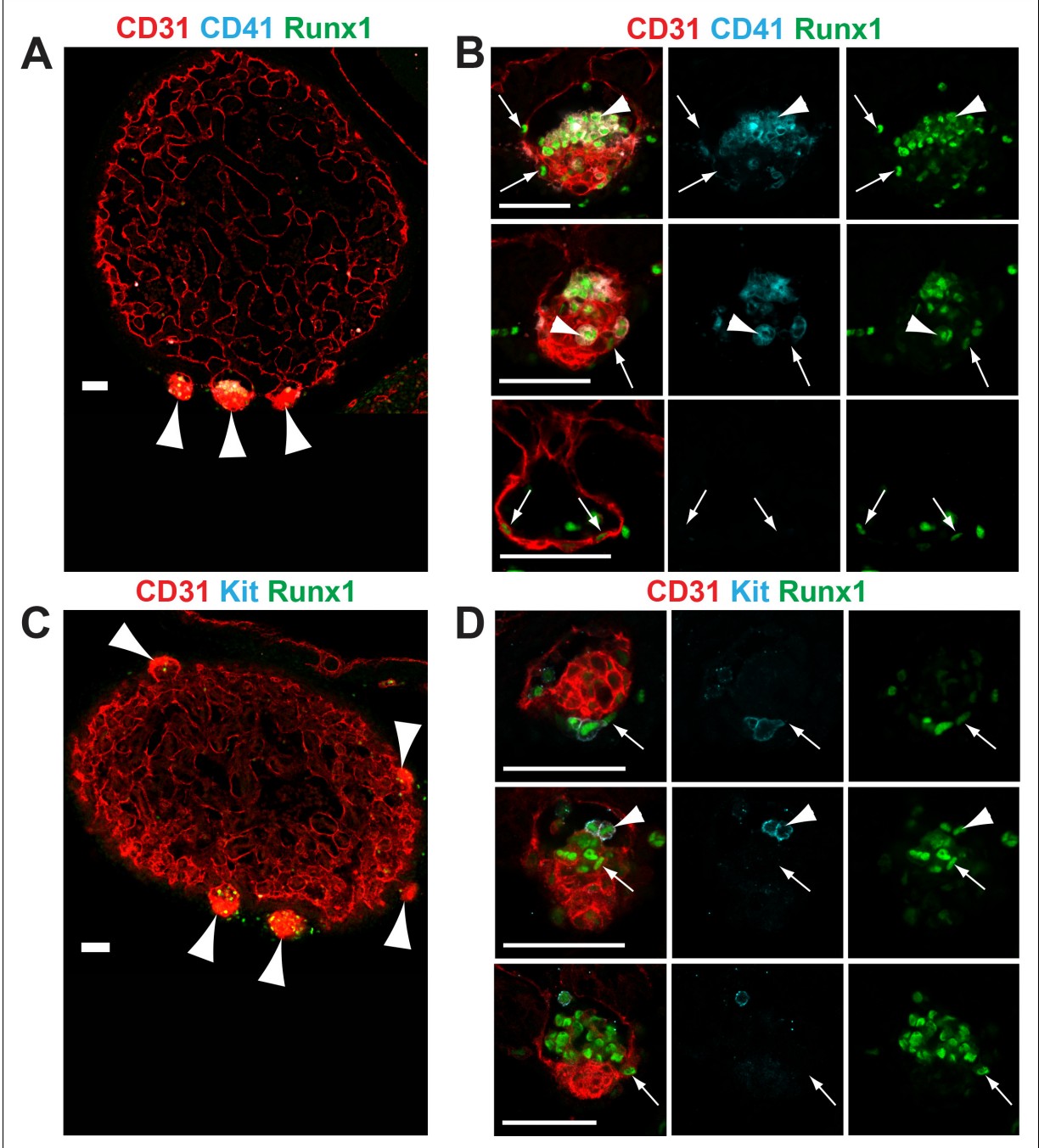

**Figure 3.** *Nf1-/-* cardiac blood islands. (**A**) Single optical projection through the ventricle of an E11.5 *Nf1-/-* embryo immunostained for CD31 (red), CD41 (cyan) and Runx1 (green). Blood islands (arrowheads) are visible sprouting from the ventricles of *Nf1-/-* embryos. (**B**) Single optical projection through blood islands on the ventricles of E11.5 *Nf1-/-* embryos. Runx1+ endocardial cells are visible in the blood islands (arrows). Arrowheads indicate examples of CD31+ CD41+ Runx1+ hematopoietic cells. (**C**) Single optical projection through the ventricle of an E11.5 *Nf1-/-* embryo immunostained for CD31 (red), Kit (cyan) and Runx1 (green). (**D**) Single optical projection through blood islands. Arrows indicate Runx1+ endocardial cells. Arrowheads indicate examples of CD31+Kit+ Runx1+ hematopoietic cells. Scale bars = 50 μm.

mutation of the conserved 'arginine finger' within the GAP domain decreases neurofibromin GAP function while leaving the domain structurally intact (*Ahmadian et al., 1997*; *Scheffzek et al., 1997*; *Klose et al., 1998*; *Hiatt et al., 2004*). These observations indicate that inactivation of neurofibromin GAP activity elevates the phosphorylation of the Ras pathway effector ERK *in vivo*.

*Nf1GRDCTL* mice either heterozygous or homozygous for the control allele in which arginine 1276 was left intact, appeared normal. Intercrosses of *Nf1GRDCTL/+* mice produced 6 of 26 *Nf1GRDCTL/GRDCTL* offspring (23%). These control mice were not examined further, and we conclude that embryonic lethality observed in *Nf1GRD/GRD* embryos is due specifically to the R1276P mutation.

## *Nf1GRD/GRD* embryos exhibit cardiac endocardial cushion and neural crest defects

Histologic analysis of E11.5 *Nf1GRD/GRD* embryos revealed abnormal cardiac outflow tract morphology and enlarged endocardial cushions, similar to those seen in *Nf1-/-* embryos (*Figure 5A*), which have been previously described in detail (*Brannan et al., 1994*; *Jacks et al., 1994*; *Lakkis and Epstein, 1998*). Atrioventricular endocardial cushions were also enlarged and ventricular septal defects were present, similar to the phenotype seen in *Nf1-/-* embryos (*Figure 5B*). Sympathetic ganglia, derived from neural crest, were enlarged in both *Nf1GRD/GRD* and *Nf1-/-* embryos (*Figure 5C*). Enlargement of sympathetic ganglia in *Nf1GRD/GRD* mutants was confirmed by immuno-fluorescence staining for neurofilament and tyrosine hydroxylase (*Figure 6A,B*).

## Neural crest-specific loss of *Nf1* Ras-GAP function leads to tissue overgrowth

Tissue-specific loss of *Nf1* in neural crest results in late gestation lethality, bypassing the midgestation cardiac defects seen in *Nf1* null mutants (*Gitler et al., 2003*). In order to examine in more detail if the Ras-GAP function of neurofibromin is necessary in developing neural crest in embryos surviving past midgestation, we crossed *Wnt1-Cre; Nf1GRD/+* mice with *Nf1flox/flox* mice to generate *Wnt1-Cre; Nf1GRD/flox* offspring. At E18.5-P0, no viable *Wnt1-Cre; Nf1GRD/flox* pups were identified out of 80 genotyped, although 12 non-viable pups (15%) were stillborn or died shortly after birth (*Table 2*). Live *Wnt1-Cre; Nf1GRD/flox* embryos were recovered between E12.5 and 16.5 at the expected frequency (*Table 2*).

Histologic examination of *Wnt1-Cre; Nf1GRD/flox* embryos revealed overgrowth of the adrenal medulla when compared to control *Nf1GRD/flox* embryos that phenocopied adrenal medullary defects seen in *Wnt1-Cre; Nf1flox/flox* embryos (*Figure 7A*), described previously (*Gitler and Epstein, 2003*). Massive enlargement of paraspinal neural crest-derived ganglia was also noted in both *Wnt1-Cre; Nf1GRD/flox* and *Wnt1-Cre; Nf1flox/flox* embryos (*Figure 7B,C*). These findings suggest that loss of neurofibromin Ras-GAP function in neural crest is sufficient to reproduce the late-gestation lethality and tissue overgrowth that results from by tissue-specific deletion of *Nf1* in neural crest.

## Ectopic cardiac blood island formation is due to loss of *NF1* Ras-GAP activity

We examined E11.5 *Nf1GRD/GRD* embryos for evidence of cardiac blood island formation to determine if this results from the loss of Ras-GAP activity. *Nf1GRD/GRD* embryos appeared grossly normal at E11.5 (*Figure 8A*), but blood-filled protrusions were often visible on the ventricles (*Figure 8B*, arrowheads). Whole-mount immunofluorescence revealed that the blood-filled protrusions were phenotypically identical to the ectopic cardiac blood islands that formed on the ventricles of *Nf1-/-* embryos (*Figure 8C*, arrowheads). Furthermore, *Nf1GRD/GRD* embryos had enlarged fetal livers populated with Runx1$^+$ CD41$^+$ hematopoietic cells, similar to *Nf1-/-* embryos (*Figure 8C*). Ventricular blood islands were evident in histologic sections after hematoxylin and eosin (H and E) staining of *Nf1GRD/GRD* and *Nf1-/-* embryos and had similar structural characteristics (*Figure 8D*, arrowheads). Single optical projections through *Nf1GRD/GRD* blood islands confirm that they were associated with CD31$^+$ CD41$^+$ Runx1$^+$ phenotypic hematopoietic cells (*Figure 8E*). An

**Table 1.** Genotypes from *Nf1GRD/+* X *Nf1GRD/+* intercrosses.

| Age | Total | +/+ | Nf1GRD/+ | Nf1GRD/GRD |
|---|---|---|---|---|
| E11.5 | 63 | 13 | 39 | 11 |
| E12.5 | 61 | 19 | 41 | 1 |
| P0 | 62 | 23 | 39 | 0 |

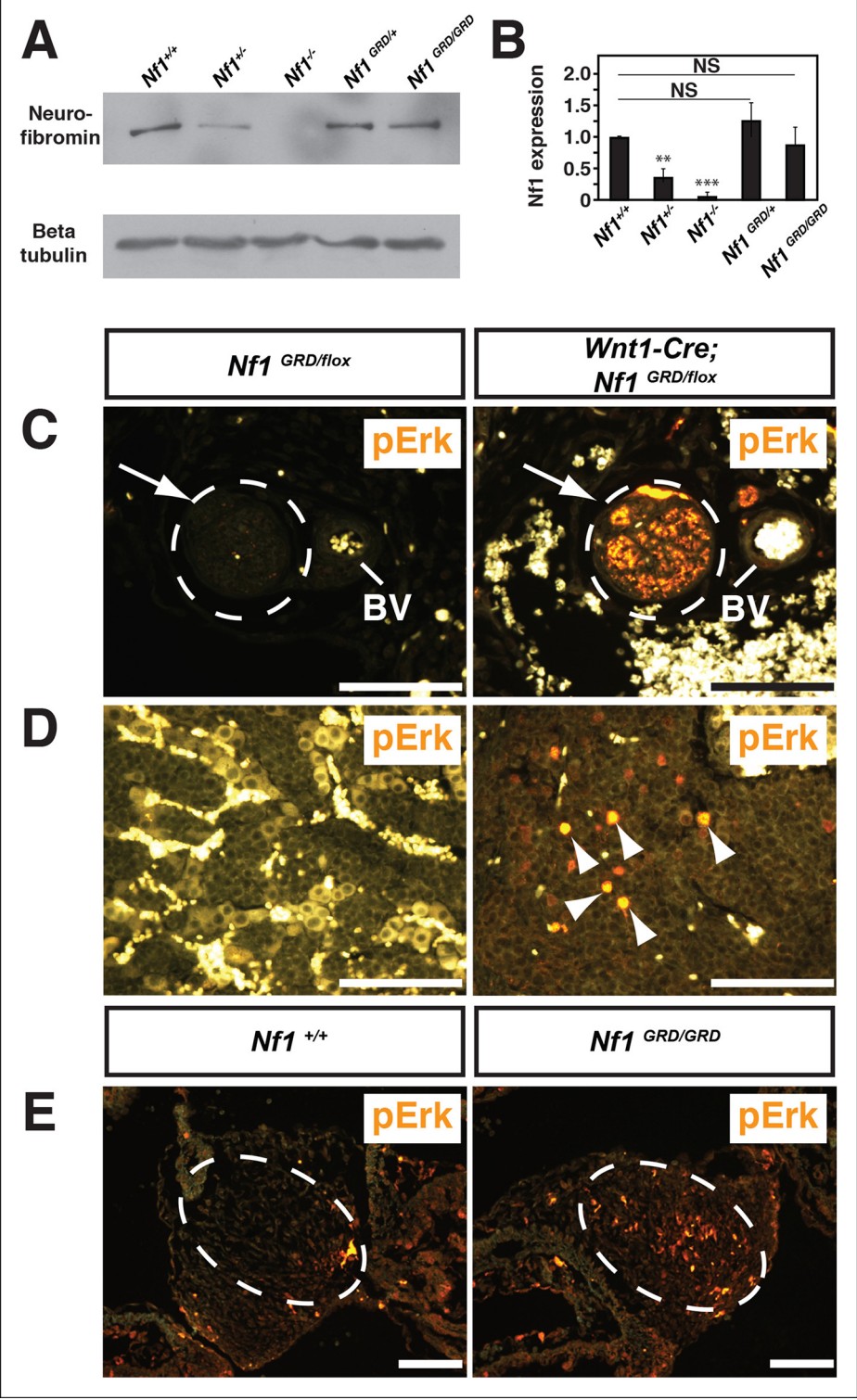

**Figure 4.** Neurofibromin protein expression and activity from the *Nf1* alleles. (**A**) Total cell lysates from E10.5 *Nf1$^{+/+}$*, *Nf1$^{+/-}$*, *Nf1-/-*, *Nf1$^{GRD/+}$*, and *Nf1$^{GRD/GRD}$* embryos were analyzed by SDS-PAGE followed by immunoblotting with either anti-neurofibromin (top panel) or anti-beta tubulin (bottom panel) antibodies as indicated. (**B**) Band intensities from 5 immunoblots as in (**A**) were quantified by ImageJ. The relative neurofibromin expression for each genotype compared to wild-type is indicated. All data are represented as the mean ± S.E. **, $p<0.05$; ***, $p<0.001$; NS = not significant ($p<0.001$, one-way ANOVA between groups, post hoc multiple comparisons, Tukey's test). (**C**) A cross-section of a peripheral nerve (demarcated in white and indicated by an arrow) from each of

*Figure 4. continued on next page*

*Figure 4. Continued*

*Nf1*$^{GRD/flox}$ and *Wnt1-Cre; Nf1*$^{GRD/flox}$ P0 animals shows elevated expression of pERK, a downstream indicator of Ras activity, in *Wnt1-Cre; Nf1*$^{GRD/flox}$ animals (right panel). An adjacent blood vessel (BV) is indicated. (**D**) Adrenal medullary tissue within an adrenal gland from either a *Nf1*$^{GRD/flox}$ or *Wnt1-Cre; Nf1*$^{GRD/flox}$ animal shows increased pERK expression in a hyperplastic area from the *Wnt1-Cre; Nf1*$^{GRD/flox}$ animal (right panel). pERK-positive cells are marked by arrowheads. Background fluorescence from non-neural-crest-derived adrenal cortical and blood cells is evident in the *Nf1*$^{GRD/flox}$ sample. (**E**) Cardiac cushions from E11.5 embryos show elevated pERK staining in *Nf1*$^{GRD/GRD}$ embryos compared to *Nf1*$^{+/+}$ animals as indicated within the dashed oval. Scale bars = 50 μm.

The following figure supplements are available for Figure 4:

**Figure supplement 1.** Generation of *Nf1*$^{GRD}$ and *Nf1*$^{GRDCTL}$ mouse lines.

**Figure supplement 2.** Increased pERK staining in neural crest derivatives of *Nf1*$^{GRD/flox}$ newborn animals following deletion by *Wnt1-Cre.*

average of 26.3 ± 9.2 blood islands could be identified via confocal microscopy on the ventricles of E11.5 *Nf1*$^{GRD/GRD}$ embryos, whereas *Nf1*$^{+/+}$ and *Nf1*$^{GRD/+}$ ventricles contained no cardiac blood islands (*Figure 8F*). However, *Nf1*-deficient E11.5 embryos had, on average, >2 fold more morphological cardiac blood islands as compared to *Nf1*$^{GRD/GRD}$ embryos (compare *Figure 8F and 2D*, p≤0.022), suggesting that the *Nf1*$^{GRD}$ is a hypomorphic *Nf1* allele, at least in regard to cardiac blood island formation. Flushed *Nf1*$^{GRD/GRD}$ ventricles contained significantly more EMPs than *Nf1*$^{+/+}$ and *Nf1*$^{GRD/+}$ littermates (*Figure 8G*), but there was a trend towards fewer progenitors than in *Nf1-/-* embryos.

## Discussion

In this study we have identifed ectopic cardiac blood island formation as a novel phenotype that arises in *Nf1*-deficient embryos. Furthermore, using a mouse that expresses a mutant form of neurofibromin with decreased Ras-GAP activity, we demonstrated that the phenotype is a direct result of dysregulation of the Ras signaling pathway. We also showed that some endocardial cells in the ectopic blood islands express Runx1, a master regulator of hematopoiesis and a marker of HE. This, in addition to the enrichment of both phenotypic and functional hematopoietic progenitors in the ventricles of E11.5 *Nf1*-deficient embryos, suggests that the endocardial cells are producing hematopoietic cells *de novo*.

We also observed dysregulation of *de novo* hematopoietic progenitor formation in *Nf1*-deficient embryos in normal sites of hematopoiesis. A previous study in zebrafish embryos found that the downstream effector of the Ras signaling pathway, pERK, has a biphasic role in blood cell formation from endothelium (*Zhang et al., 2014*). When zebrafish embryos were treated with an ERK signaling inhibitor prior to artery-—vein specification, *runx1* and *myb* expression in the dorsal aorta decreased; however, when treated with an ERK signaling inhibitor after artery-—vein specification but before EHT, *runx1 and myb* expression increased (*Zhang et al., 2014*). Thus, early in development pERK is necessary for *de novo* generation of hematopoietic cells, but after artery-vein specification, pERK inhibits the specification of HE cells (*Zhang et al., 2014*). Consistent with a role for ERK signaling in HE specification, increased signaling through the fibroblast growth factor (FGF) receptor, which is upstream of ERK and regulated by Ras-GAP, decreases *runx1* expression in the dorsal aorta of zebrafish (*Pouget et al., 2014*). The mechanism by which increased FGF signaling decreases *runx1* expression in the HE involves inhibition of bone morphogenic protein signaling, which is required for *runx1* expression (*Pouget et al., 2014*; *Wilkinson et al., 2009*; *Pimanda et al., 2007*). Consistent with these findings, we show that loss of *Nf1*, which is associated with activation of the Ras-pERK pathway, results in fewer Runx1$^+$ HE cells in the dorsal aorta at E10.5, as well as fewer CD31$^+$ Kit$^+$ Runx1$^+$ hematopoietic cluster cells. In contrast, the yolk sac of *Nf1-/-* embryos produced more EMPs when compared to littermate controls, consistent with the positive role for FGF signaling in regulating erythropoiesis and myelopoiesis at a similar earlier stage in the zebrafish embryo (*Yamauchi et al., 2006*; *Walmsley et al., 2008*). Thus the level of Ras activity must be

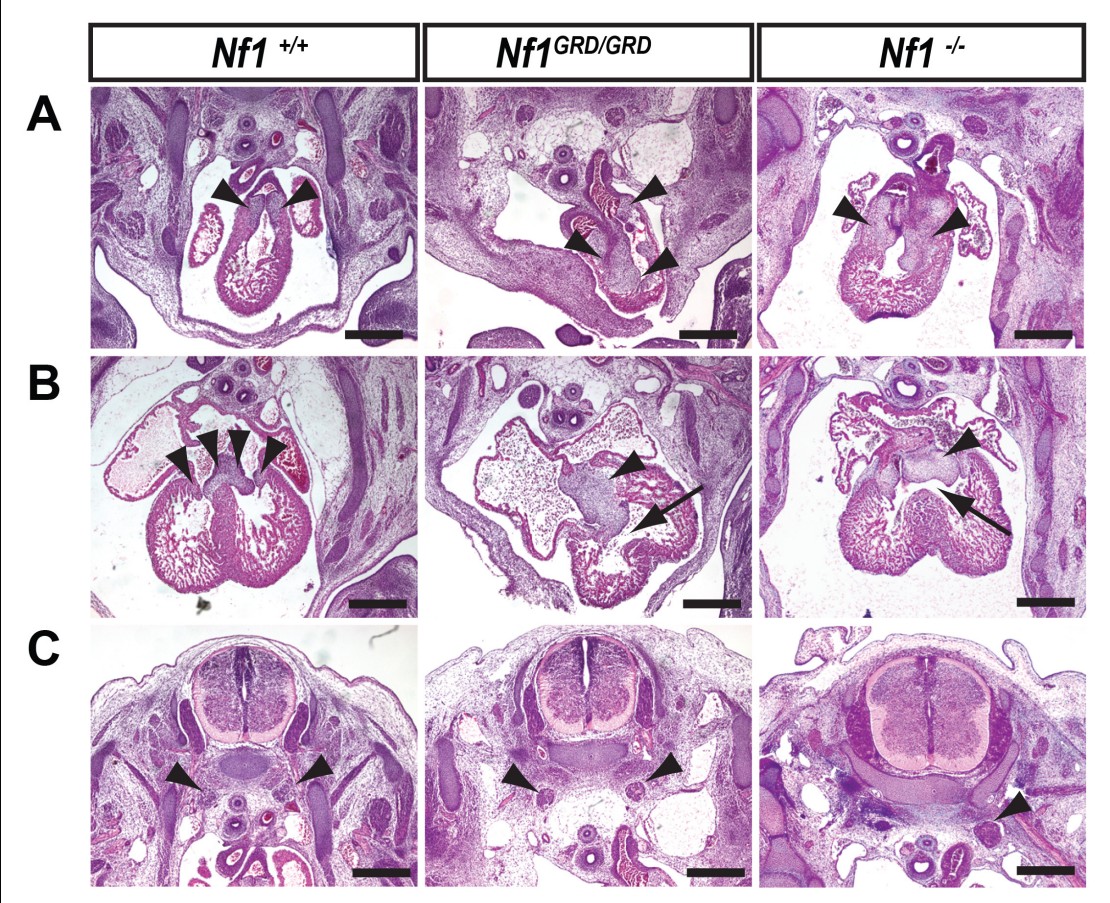

**Figure 5.** Inactivation of *Nf1* GRD function affects heart and sympathetic ganglia development. (A) Sections of hearts from E12.5-13.5 *Nf1$^{+/+}$*, *Nf1$^{GRD/GRD}$*, and *Nf1-/-* embryos. The enlarged endocardial cushions in hearts from *Nf1$^{GRD/GRD}$* embryos (arrowheads) are similar to the oversized cushions of *Nf1-/-* embryos. (B) Enlarged atrioventricular endocardial cushions (arrowheads) and ventricular septa defects (arrows) in *Nf1$^{GRD/GRD}$* and *Nf1-/-* embryos. (C) Sympathetic ganglia (arrowheads) are similarly enlarged in *Nf1$^{GRD/GRD}$* and *Nf1-/-* embryos. Scale bars = 500 μm.

carefully titrated, as elevating Ras signaling in the dorsal aorta limits HE specification, but enhances EMP formation in the yolk sac, and unleashes the hematopoietic potential of the endocardium.

It was previously reported that hematopoietic cells derived from the fetal livers of *Nf1*-deficient mice are hyperproliferative and cause a JMML-like myeloid proliferative disorder when transplanted into irradiated recipients (*Birnbaum et al., 2000*; *Largaespada et al., 1996*; *Zhang et al., 2001*; *Zhang et al., 1998*). Based on immunoflourescence, it appears that hematopoiesis is elevated as early as E11.5 in fetal livers of both *Nf1-/-* and *Nf1$^{GRD/GRD}$* embryos compared to controls, thus implicating activated Ras in hyperproliferation of hematopoietic cells (at E11.5, primarily EMPs) that populate the fetal liver. Likewise, our results implicate the loss of neurofibromin Ras-GAP function within neural crest cells as sufficient to result in overgrowth of sympathetic and dorsal root ganglia and of the adrenal medulla.

The ability of the *Nf1* gene product to act as a Ras GAP has been known for a quarter of a century (*Ballester et al., 1990*; *Xu et al., 1990*), but the degree to which this function accounts for some or all *Nf1* phenotypes has been an ongoing topic of research with relevance for therapeutic strategies. We and others have provided evidence for the ability of neurofibromin to affect alternate signaling pathways, including PKA and cAMP (*Guo et al., 1997*; *The et al., 1997*; *Hegedus et al., 2007*; *Brown et al., 2010*; *Wolman et al., 2014*). Prior work has suggested that midgestation embryonic lethality resulting from loss of *Nf1* can be rescued by transgenic expression of the isolated neurofibromin GRD, but this was not sufficient for rescue of neural crest overgrowth (*Ismat et al., 2006*). Failure to rescue neural crest overgrowth could have been the result of

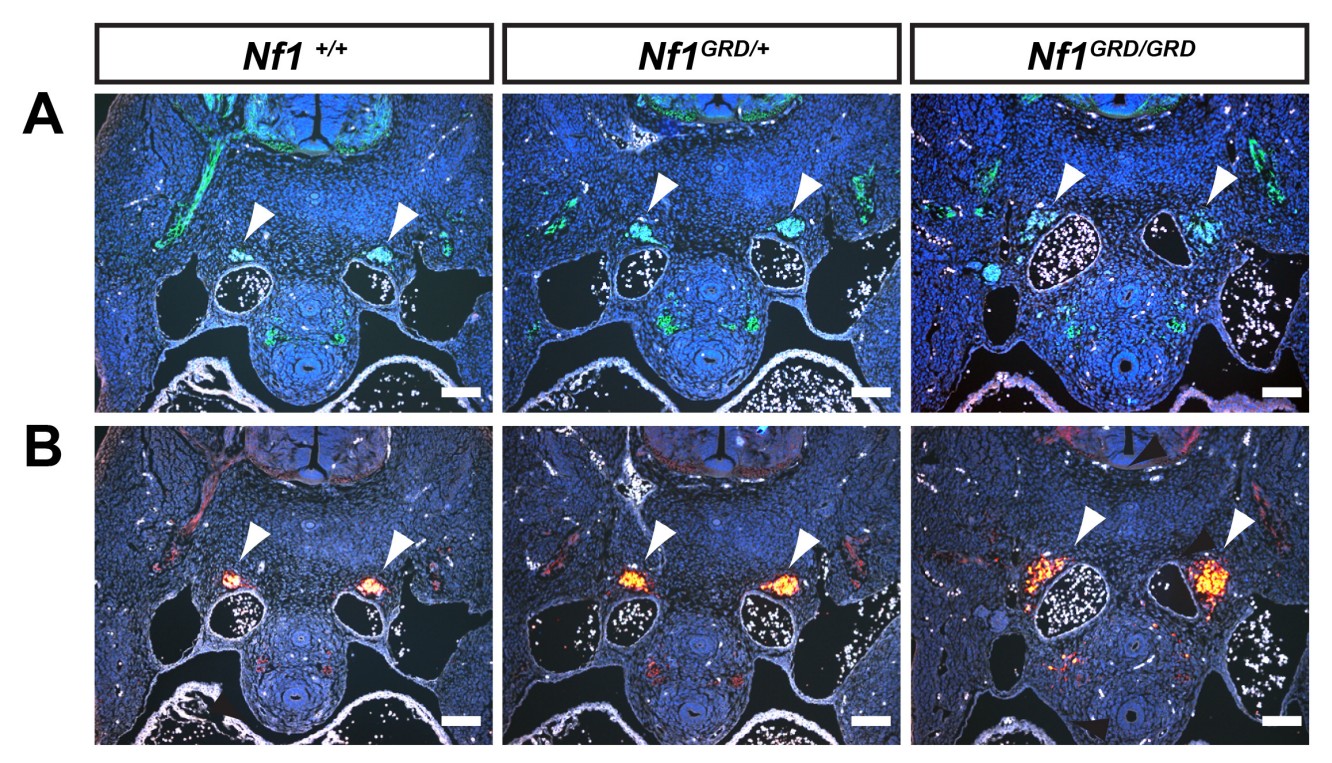

**Figure 6.** Enlarged sympathetic ganglia in E11.5 *Nf1*$^{GRD/GRD}$ embryos. (**A**) Transverse sections of E11.5 *Nf1*$^{+/+}$, *Nf1*$^{GRD/+}$, and *Nf1*$^{GRD/GRD}$ embryos stained with antibodies against neurofilament. Arrowheads indicate sympathetic ganglia. (**B**) Transverse sections of E11.5 *Nf1*$^{+/+}$, *Nf1*$^{GRD/+}$, and *Nf1*$^{GRD/GRD}$ embryos stained with antibodies against tyrosine hydroxylase. Scale bars = 100 μm.

inadequate transgenic expression of GRD in this tissue, or because of the necessity for an additional function of neurofibromin outside of the GRD. The findings reported here for *Nf1*$^{GRD/GRD}$ embryos do not rule out the existence of critical non-GRD functions of neurofibromin in the neural crest or other tissues. In fact, the *Nf1*$^{GRD}$ has characteristics of a hypomorphic allele that could be explained by non-GRD related functions. Rather, we demonstrate the necessity of GRD function for normal embryonic development. The development of the *Nf1*$^{GRD/+}$ mouse line described here will allow researchers to determine the necessity of GRD function across the spectrum of developmental and tumor phenotypes observed in mouse models of neurofibromatosis.

## Materials and methods

### Hematopoietic progenitor assay

Embryos were removed from the uterus and dissected in phosphate buffered saline (PBS) with 20% fetal bovine serum and antibiotics. The yolk sacs were removed and the hearts were dissected, the atrium was cut away and the ventricles were flushed with PBS using an insulin needle and syringe to

**Table 2.** Genotypes from *Wnt1-Cre; Nf1*$^{GRD/+}$ X *Nf1*$^{flox/flox}$ crosses.

| Age | Total | Nf1$^{flox/+}$ | Nf1$^{GRD/flox}$ | Wnt1-Cre;Nf1$^{flox/+}$ | Wnt1-Cre;Nf1$^{GRD/flox}$ |
|---|---|---|---|---|---|
| E12.5-–16.5 | 27 | 10 | 5 | 3 | 7 * |
| E18.5-P0 | 80 | 32 | 18 | 18 | 0 ** |

*2 non-viable *Wnt1-Cre; Nf1*$^{GRD/flox}$ embryos were recovered at E12.5-–16.5
**12 non-viable *Wnt1-Cre; Nf1*$^{GRD/flox}$ pups were recovered at E18.5-P

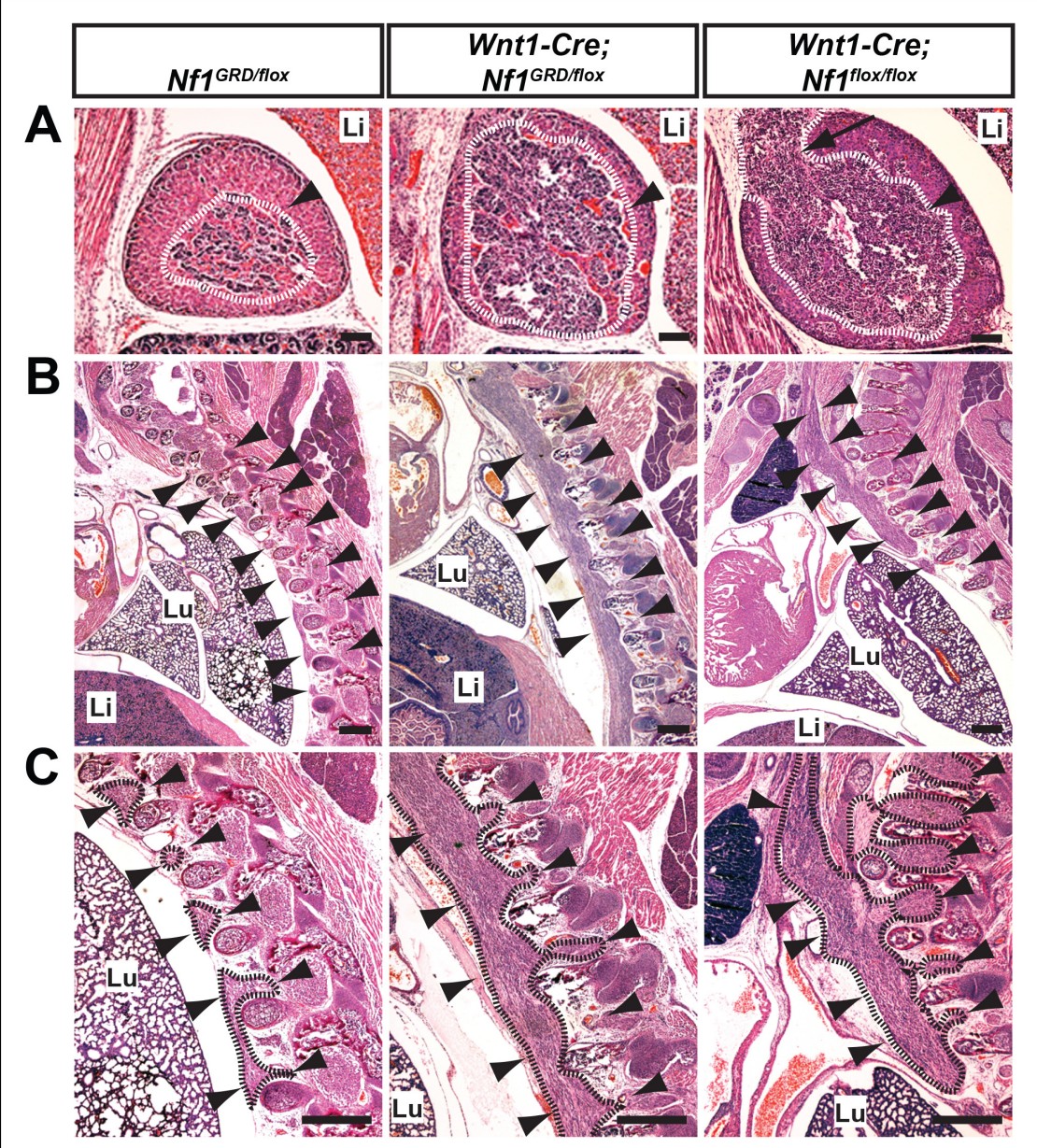

**Figure 7.** Hyperplasia of neural crest derivatives is similar in $Nf1^{GRD/flox}$ and $Nf1^{flox/flox}$ newborn animals in which $Nf1$ is deleted in neural crest cells with Wnt1-Cre. (**A**) Adrenal medullary tissue (demarcated in white and indicated with an arrowhead) contained within an adrenal gland of P0 wild-type, P0 Wnt1-Cre; $Nf1^{GRD/flox}$, or E16.5 Wnt1-Cre; $Nf1^{flox/flox}$ animals. The tissue is similarly overgrown in Wnt1-Cre; $Nf1^{GRD/flox}$ and Wnt1-Cre; $Nf1^{flox/flox}$ newborns/fetuses. Scale bars = 100 μm. The arrow indicates a tumor-like medullary protrusion. (**B**) Sagittal sections showing peripheral ganglia (arrowheads) in $Nf1^{GRD/flox}$ newborn pups, and abnormally enlarged ganglia and tumor-like overgrowth of nerve tissue adjacent to the lumbar spine in a Wnt1-Cre; $Nf1^{GRD/flox}$ newborn pup and an E16.5 Wnt1-Cre; $Nf1^{flox/flox}$ fetuse. Scale bars = 500 μm. (**C**) Magnifications of images in (**B**), with hyperplastic tissue demarcated in black and marked by arrowheads. Lu, lung; Li, liver; Scale bars = 500 μm.

remove circulating blood cells. The ventricles and yolk sacs were then dissociated in 0.125% collagenase Type I (Sigma, St Louis, MO) for 20–30 min at 37°C, triturated, washed, and filtered to obtain a single cell suspension. Single cell solutions of embryonic ventricles or yolk sacs were plated in methylcellulose (MethocultM3434; Stem Cell Technologies, Vancouver, BC) and colonies were counted 7–8 days after plating.

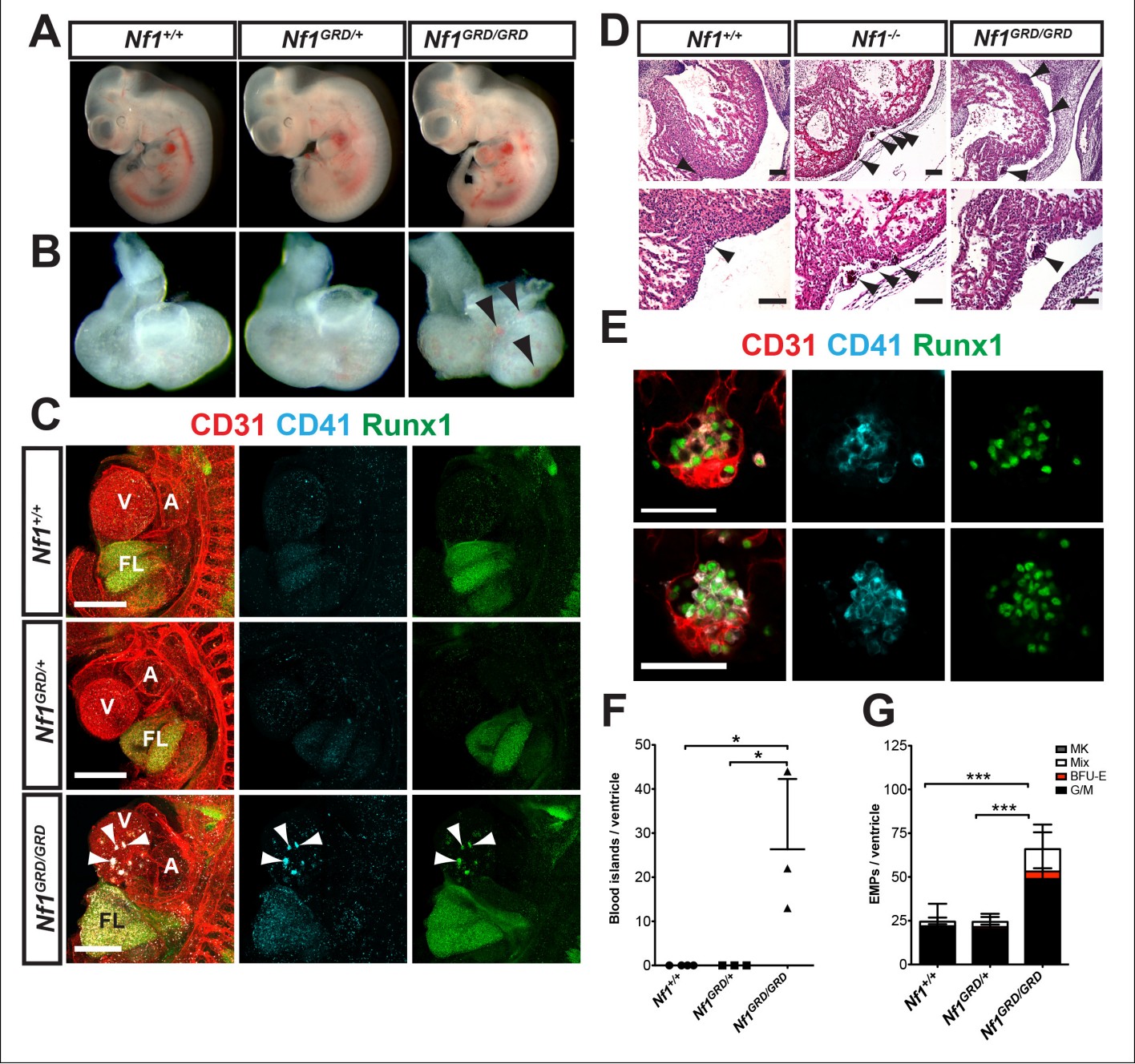

**Figure 8.** E11.5 *Nf1^GRD/GRD* embryos form ectopic cardiac blood islands. (**A**) Gross view of E11.5 *Nf1^+/+*, *Nf1^GRD/+* and *Nf1^GRD/GRD* littermates. (**B**) Isolated hearts from E11.5 *Nf1^+/+*, *Nf1^GRD/+* and *Nf1^GRD/GRD* embryos. Black arrowheads indicate blood-filled protrusions on the ventricle of the *Nf1^GRD/GRD* heart. (**C**) Confocal Z-projections (Z interval = 5 μm) of CD31 (red), CD41 (cyan) and Runx1 (green) immunostained E11.5 *Nf1^+/+*, *Nf1^GRD/+* and *Nf1^GRD/GRD* embryos. Arrowheads point to blood islands on the ventricle of the *Nf1^GRD/GRD* embryo. Scale bars = 500 μm. (**D**) Cell aggregates resembling blood islands (arrowheads) in hearts of E12.5 *Nf1^+/+*, *Nf1-/-* and *Nf1^GRD/GRD* embryos. Lower panels, are magnifications of images in top panels. Scale bars = 100 μm. (**E**) Single optical projection through the cardiac blood islands of an E11.5 *Nf1^GRD/GRD* embryo immunostained for CD31 (red), Runx1 (green) and CD41 (cyan). Scale bars = 50 μm. (**F**) Quantification of blood islands on the ventricles of E11.5 embryos. One-way ANOVA and Bonferroni's multiple comparison test applied to determine significance, error bars represent the SD. (**G**) Number of erythroid and myeloid progenitors per flushed E11.5 ventricle. One-way ANOVA and Bonferroni's multiple comparison test applied to determine significance; error bars represent SD. *Nf1^+/+* n = 21, *Nf1^GRD/+* n = 22 and *Nf1^GRD/GRD* n = 8. * indicates that $p \leq 0.05$ and *** indicates that $p \leq 0.001$. V: ventricle; A: atrium; FL: fetal liver; Mk: megakaryocyte; Mix: granulocyte-erythroid-monocyte-megakaryocyte; BFU-E: burst forming unit-erythroid; G/M: granulocyte-macrophage colonies.

## Whole-mount immunofluorescence and confocal microscopy

Embryos were prepared as described previously (*Yokomizo et al., 2012*). The following primary antibodies were used; rat anti-mouse CD31 (Mec 13.3, BD Pharmingen, San Diego, CA), rat anti-mouse CD117 (2B8, eBiosciences, San Diego, CA), rat anti-mouse CD41 (MWReg30, BDBiosciences, Franklin Lakes, NJ) and rabbit anti-human/mouse Runx (EPR3099, Abcam, Cambridge, MA). Secondary antibodies used were goat anti-rat Alexa Fluor 647 (Invitrogen, Carlsbad, CA), goat-anti rat Alexa Fluor 555 (Abcam) and goat anti-rabbit Alexa Fluor 488 (Invitrogen). Images were acquired on a Zeiss LSM 710 AxioObserver inverted microscope with ZEN 2011 software and processed with Fiji software (*Schindelin, et al., 2012*). Hematopoietic cells in the dorsal aorta were counted using the cell counter plugin (version February 29, 2008, Kurt De Vos; http://rsb.info.nih.gov/ij/plugins/cell-counter.html).

## Immunoblotting

E10.5 embryos were dissected free of the amniotic sac, frozen in liquid nitrogen, thawed, and disrupted by pipetting in Hank's Balanced Salt Solution containing 5 mM ethylenediaminetetraacetic acid (EDTA). Total cell lysates were prepared by heating samples in boiling Laemli buffer (66 mM Tris–HCl, pH 6.8, 2% (w/v) SDS, 10 mM EDTA). The samples were subjected to sodium dodecyl sulfate polyacrylamide gel electrophoresis (SDS-PAGE) and immunoblotting analysis using anti-neurofibromin antibody ab17963 (Abcam). Immunoreactive bands were visualized by chemiluminescence. Quantification of individual band intensities was performed using ImageJ. One-way analysis of variance (ANOVA) was used to assess statistical differences between band intensities. Significant ANOVA results were analyzed post hoc by the Tukey-Kramer multiple comparisons test.

## Histology and immunofluorescence analyses

Whole mouse embryos or dissected hearts were fixed in 2 or 4% paraformaldehyde, dehydrated in ethanol, and embedded in paraffin for sectioning. Tissues were visualized with H and E stain or by immunofluorescent detection of marker proteins according to standard practices. Detailed protocols are available at http://www.pennmedicine.org/heart/. Antibodies used for immunofluorescence include rabbit polyclonal anti-tyrosine hydroxylase (AB152, EMD Millipore/Chemicon, Billerica, MA), rabbit polyclonal anti-pERK (#9101, Cell Signaling Technology, Inc., Danvers, MA) and mouse monoclonal anti-neurofilament (2H3, Developmental Studies Hybridoma Bank, Department of Biology, University of Iowa, Iowa City, IA). Images were adjusted using Adobe Photoshop using settings applied across the entirety of each image.

## Mice

All mouse manipulations were performed in accordance with protocols approved by the Institutional Animal Care and Use Committee (IACUC) of the University of Pennsylvania following guidelines described in the US National Institutes of Health *Guide for the Care and Use of Laboratory Animals.*

Nf1-/-, Nf1$^{flox/+}$ and Wnt1-Cre mice have been described previously (*Brannan et al., 1994*; *Jacks et al., 1994*; *Zhu et al., 2001*; *Danielian et al., 1998*; *Jiang et al., 2000*). Nf1$^{GRD/+}$ and Nf1$^{GRDCTL/+}$ mice were produced by targeting C57BL/6 ES cells (Genoway, Lyon, France) with a targeting vector designed to replace arginine 1276 with proline (R1276P) or, in the case of Nf1$^{GRDCTL/+}$ to leave arginine 1276 as arginine. The selection strategy (*Figure 4—figure supplement 1*) included a self-excising floxed neomycin resistance cassette that, after excision, leaves a single loxP site within intron 27. The Nf1$^{GRDCTL/+}$ mice were created in order to control for possible unpredicted effects related to the introduction of small changes in genomic sequence, other than those encoding the R1276P missense mutation, necessitated by the targeting strategy. Nf1$^{GRD/+}$ and Nf1$^{GRDCTL/+}$ mice were genotyped using polymerase chain reaction primers listed below, which produce a 175 bp wild-type band and a 248 bp mutant band (*Figure 4—figure supplement 1*). All mice were maintained on a C57BL/6 background.

GRDF: 5'- GAGGGGAGATGTCAAAGATGTATTGTGTAACTAC-3'
GRDR: 5'- CAACCTTCAAACAGTACTAAAGTCCATCATGG-3'

## Acknowledgements

Core services were provided by the Abramson Family Cancer Research Institute, the Abramson Cancer Center and the Histology and Gene Expression Core of the Penn Cardiovascular Institute. We especially thank Andrea Stout and Jasmine Zhao at the UPenn Cell & Developmental Biology Microscopy Core for confocal microscopy assistance and Haig Aghajanian of the Epstein lab for assistance with gel band quantification and statistical analysis.

## Additional information

### Funding

| Funder | Grant reference number | Author |
| --- | --- | --- |
| National Heart, Lung, and Blood Institute | R01HL091724 | Nancy A Speck |
| Cotswold Foundation | | Jonathan A Epstein |
| Spain Fund | | Jonathan A Epstein |
| National Heart, Lung, and Blood Institute | U01HL100405 | Nancy A Speck Jonathan A Epstein |
| National Heart, Lung, and Blood Institute | 1F31HL120615 | Amanda D Yzaguirre |
| National Cancer Institute | T32CA09140 | Amanda D Yzaguirre |
| WW Smith Endowed Chair | | Jonathan A Epstein |

The funders had no role in study design, data collection and interpretation, or the decision to submit the work for publication.

### Author contributions

ADY, AP, EDdeG, KAE, Conception and design, Acquisition of data, Analysis and interpretation of data, Drafting or revising the article; JL, Conception and design, Acquisition of data, Drafting or revising the article; NAS, JAE, Conception and design, Analysis and interpretation of data, Drafting or revising the article

### Ethics

Animal experimentation: This study was performed in strict accordance with the recommendations in the Guide for the Care and Use of Laboratory Animals of the National Institutes of Health. All of the animals were handled according to approved institutional animal care and use committee (IACUC) protocols (#803789 and #803317) of the University of Pennsylvania.

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
