## [Decision Letter]

Thank you for submitting your work entitled "Loss of Ras-GAP activity of neurofibromin enhances formation of cardiac blood islands in murine embryos" for peer review at *eLife*. Your submission has been favorably evaluated by Sean Morrison (Senior editor) and three reviewers, one of whom served as a guest Reviewing editor.

The reviewers have discussed the reviews with one another and the Reviewing editor has drafted this decision to help you prepare a revised submission.

This paper shows that germline inactivation of the *Nf1* tumor suppressor gene leads to aberrant formation of cardiac blood islands, which are observed during a discrete developmental window. This study also describes a novel knock in *Nf1* mutant allele containing an amino acid substitution in the catalytic arginine residue that interacts with the P loop of Ras proteins. This conditional "GRD" knock in allele failed to rescue developmental phenotypes associated with *Nf1* inactivation including aberrant blood island formation. The analyses of blood islands and of hematopoietic cluster cells that appear earlier in development extend our knowledge regarding the role of *Nf1* in hematopoietic development. These observations are interesting and relevant given the association of NF1 with JMML and the finding of somatic NF1 gene mutations in other blood cancers. The GRD knock in allele is a novel and valuable resource that will facilitate additional studies on the role of NF1 in development, neuro-cognitive abnormalities, and tumorigenesis.

Essential revisions:

1) The finding that disruption of NF1 increases blood island formation and blood progenitor development highlights the hemogenic capacity in the endocardium. Given this, a more complete analysis of endocardial hematopoiesis in these embryos is warranted. The assays reported in the manuscript are limited to immunofluorescence for hemogenic endothelial markers and colony assays, which suggest that NF1 deficient endocardial cells are at least capable of erythro-myeloid (EMP) progenitor production, which is similar to the second wave of hematopoiesis in the yolk sac. A general FACS assessment for the key cell types of embryonic hematopoietic progenitor/stem cell hierarchy in flushed hearts (e.g. primitive erythroid, EMPs and HSCs) should be well within the expertise of the team.

2) A critical missing link in the paper is that yolk sac hematopoiesis was not assessed. Is the increased EMP production limited to the heart, or also observed in the yolk sac, which is the classical site of EMP production? This would be important conceptually as it would reveal whether the increased endocardial hematopoiesis upon NF1 inactivation reflects unique regulation in the heart, or indicates a broader requirement of NF1 in hemogenic vessels that generate EMPs. FACS quantification of hematopoietic stem and progenitor cells in the key hematopoietic organs should be performed to serve as a point of reference to compare hematopoiesis in the hearts of NF1 mutant embryos.

3) There is no biochemical proof shown that the knock in mutation results in Ras hyperactivation. Some data assessing evidence of biochemical activation of Ras – perhaps in fetal liver cells from GRD/GRD mice – should be provided.

4) Quantitative data comparing blood island formation in *Nf1* WT versus GRD/ + versus GRD/GRD for multiple embryos of each genotype should be provided (similar to panel 2D). The question here (and in #5 below) is whether the GRD allele is a true null or a hypomorph.

5) The immunoblot data shown in panel 4D are central to the conclusions of the paper as they address the nature of the GRD allele. In particular, it appears that neurofibromin expression in the GRD/ + and GRD/GRD lanes are about equal to the *Nf1 ^+ /-^* lane and less that the wild-type. This panel is somewhat problematic as a loading control is not included. This is a key point as the data as presented do not exclude the possibility that wild-type expression levels of the GRD mutant might retain some biologic activity. It is essential that the authors address this point carefully through orthogonal approaches that might include quantitative RT PCR to assess GRDand GRDCTL mRNA expression and more quantitative Western blotting with appropriate loading controls to assess the expression levels of the respective GRD proteins relative to wild-type *Nf1*.

---

## [Author Response]

*1) The finding that disruption of NF1 increases blood island formation and blood progenitor development highlights the hemogenic capacity in the endocardium. Given this, a more complete analysis of endocardial hematopoiesis in these embryos is warranted. The assays reported in the manuscript are limited to immunofluorescence for hemogenic endothelial markers and colony assays, which suggest that NF1 deficient endocardial cells are at least capable of erythro-myeloid (EMP) progenitor production, which is similar to the second wave of hematopoiesis in the yolk sac. A general FACS assessment for the key cell types of embryonic hematopoietic progenitor/stem cell hierarchy in flushed hearts (e.g. primitive erythroid, EMPs and HSCs) should be well within the expertise of the team.* Wild type embryonic hearts contain too few progenitors for FACS analysis. For example, Figure 2 shows that each normal embryonic heart contains approximately 25 CFU-C progenitors. At E11.5 there are approximately 3-4 HSCs per embryo, and no good markers to distinguish them from committed progenitors. Therefore, although it is well within our expertise to analyze progenitors from more abundant tissue sources, or in situations where many embryos can be pooled for analysis, we are unable to do so in this situation.

*2) A critical missing link in the paper is that yolk sac hematopoiesis was not assessed. Is the increased EMP production limited to the heart, or also observed in the yolk sac, which is the classical site of EMP production? This would be important conceptually as it would reveal whether the increased endocardial hematopoiesis upon NF1 inactivation reflects unique regulation in the heart, or indicates a broader requirement of NF1 in hemogenic vessels that generate EMPs. FACS quantification of hematopoietic stem and progenitor cells in the key hematopoietic organs should be performed to serve as a point of reference to compare hematopoiesis in the hearts of NF1 mutant embryos.*

We analyzed EMPs in *Nf1^-/-^* yolk sacs and found that they were increased relative to wild type embryos, particularly erythroid progenitors. The difference between yolk sac and heart versus dorsal aorta is quite interesting, and indicates that Ras signaling regulates aortic hematopoiesis quite differently than yolk sac or heart hematopoiesis. The new yolk sac data are included in revised Figure 1.

*3) There is no biochemical proof shown that the knock in mutation results in Ras hyperactivation. Some data assessing evidence of biochemical activation of Ras – perhaps in fetal liver cells from GRD/GRD mice – should be provided.*

Using immunofluorescence we demonstrate that tissues from Nf1 GRD animals harboring the R1276P mutation within the GAP domain have elevated levels of the Ras effector pERK (revised Figure 4). This is consistent with multiple reports showing that mutation of the conserved “arginine finger” within the GAP domain decreases neurofibromin GAP function while leaving the domain structurally intact (Ahmadian, 1997; Scheffzek, 1997; Klose, 1998; Hiatt, K., 2001). The R1276P mutation has been utilized in multiple contexts to probe the function of neurofibromin Ras-GAP activity (Yang, FC, 2003; Hiatt, K., 2004; Hannan et al., 2006; Ho, 2007; Shannon, 2014). Just as others have found the R1276P mutation within the GAP domain a useful research tool to probe neurofibromin GAP function, we feel the *Nf1* GRD mouse will be valuable in extending these earlier observations within the setting of a mouse line.

*4) Quantitative data comparing blood island formation in* Nf1 *WT versus GRD/ + versus GRD/GRD for multiple embryos of each genotype should be provided (similar to panel 2D). The question here (and in #5 below) is whether the GRD allele is a true null or a hypomorph.*

This was an excellent suggestion, and we performed the analysis requested. Indeed *Nf1^GRD/GRD^*E11.5 embryos have fewer blood islands than *Nf1^-/-^* embryos. These new *Nf1^GRD/GRD^*data are included in Figure 8.

*5) The immunoblot data shown in panel 4D are central to the conclusions of the paper as they address the nature of the GRD allele. In particular, it appears that neurofibromin expression in the GRD/ + and GRD/GRD lanes are about equal to the* Nf1 ^+ /-^
*lane and less that the wild-type. This panel is somewhat problematic as a loading control is not included. This is a key point as the data as presented do not exclude the possibility that wild-type expression levels of the GRD mutant might retain some biologic activity. It is essential that the authors address this point carefully through orthogonal approaches that might include quantitative RT PCR to assess GRDand GRDCTL mRNA expression and more quantitative Western blotting with appropriate loading controls to assess the expression levels of the respective GRD proteins relative to wild-type* Nf1.

We have more carefully quantified neurofibromin protein expression by performing 5 independent Westerns with loading controls. A representative Western and the averaged, normalized data is shown in revised Figure 4.